# Trivariate Joint Distribution Modelling of Compound Events Using the Nonparametric D-Vine Copula Developed Based on a Bernstein and Beta Kernel Copula Density Framework

**Shahid Latif and Slobodan P. Simonovic ***

Department of Civil and Environmental Engineering, Western University, London, ON N6A 5B8, Canada
* Correspondence: simonovic@uwo.ca

**Abstract:** Low-lying coastal communities are often threatened by compound flooding (CF), which can be determined through the joint occurrence of storm surges, rainfall and river discharge, either successively or in close succession. The trivariate distribution can demonstrate the risk of the compound phenomenon more realistically, rather than considering each contributing factor independently or in pairwise dependency relations. Recently, the vine copula has been recognized as a highly flexible approach to constructing a higher-dimensional joint density framework. In these, the parametric class copula with parametric univariate marginals is often involved. Its incorporation can lead to a lack of flexibility due to parametric functions that have prior distribution assumptions about their univariate marginal and/or copula joint density. This study introduces the vine copula approach in a nonparametric setting by introducing Bernstein and Beta kernel copula density in establishing trivariate flood dependence. The proposed model was applied to 46 years of flood characteristics collected on the west coast of Canada. The univariate flood marginal distribution was modelled using nonparametric kernel density estimation (KDE). The 2D Bernstein estimator and beta kernel copula estimator were tested independently in capturing pairwise dependencies to establish D-vine structure in a stage-wise nesting approach in three alternative ways, each by permutating the location of the conditioning variable. The best-fitted vine structure was selected using goodness-of-fit (GOF) test statistics. The performance of the nonparametric vine approach was also compared with those of vines constructed with a parametric and semiparametric fitting procedure. Investigation revealed that the D-vine copula constructed using a Bernstein copula with normal KDE marginals performed well nonparametrically in capturing the dependence of the compound events. Finally, the derived nonparametric model was used in the estimation of trivariate joint return periods, and further employed in estimating failure probability statistics.

**Keywords:** compound flooding; D-vine copula; trivariate joint analysis; Bernstein estimator; beta kernel estimator; parametric copulas; kernel density estimation; return periods



## 1. Introduction

Compound events (CE) is a multidimensional phenomenon that can be defined by the joint probability occurrence of two or more extreme or non-extreme events, which may not be dangerous or devastating if considered individually [1–4]. However, CE can have severe consequences if their underlying variables co-occur or are in close succession. On the global scale, the flooding events in low-lying coastal cities or the risk of extreme compound phenomena have already been recorded and outlined in the previous literature [5–7]. Climate change has already triggered a rising coastal water level called sea level rise (SLR), increasing the frequency and severity of flooding, which threatens coastal communities worldwide [8–10]. Coastal flooding can be significantly defined and estimated by combining the driving forces, such as storm surge (oceanographic), rainfall (or pluvial flooding) and river discharge (or fluvial flooding). These events can be interlinked through a common forcing mechanism, such as tropical or extra-tropical cyclones

(or a low-atmospheric-pressure scenario). Among the different coastal flood drivers, a storm surge event is often considered a significant flood-driving agent [11]. When combined with rainfall (e.g., [2]) or with high river discharge (e.g., [10]), it can result in a devastating situation.

Different mathematical or statistical frameworks are often pointed out in the demonstration of the compound phenomenon but still can lack a consistent or robust approach. The traditional statistical evaluation of the CEs is usually a multivariate framework that observes the number of extreme joint episodes by targeting the most justifiable flood drivers. For instance, take the studies performed by Coles [12], Coles et al. [13], Svensson and Jones [14], Cooley et al. [15], Zheng et al. [16] and Zheng et al. [17]. In reality, the validity of the univariate probability or frequency analysis (and return periods) is questionable. Due to the multidimensional behaviour, it must demand an efficient framework that can reduce the hydrologic risk much more efficiently.

In recent studies, copula functions gained more popularity than traditional multivariate models and are recognized as highly flexible tools in the bivariate or multivariate joint distribution analysis of hydro-meteorological observations [18–22]. In the modelling of CE or flooding, the adequacy of different parametric class 2D copulas is tested by targeting different contributing variables, for instance, storm surge (or storm tide) and rainfall, or storm surge (or storm tide) and river discharge [23–28]. Such incorporations are limited to bivariate joint cases employing 2D parametric class copulas to observe pairwise joint dependencies. However, the more realistic and practical flood risk can be obtained by compounding the joint distribution behaviour, including more relevant flood-driving agents (e.g., storm surge, rainfall, and river discharge) simultaneously instead of their pairwise dependencies. For instance, tropical cyclones in the coastal region can trigger storm surges, rainfall and possible high-river-discharge events simultaneously; thus, the complex interplay between them can exacerbate flooding in the coastal zones. Therefore, the risk of coastal flooding can be analysed much more efficiently by considering the above triplet variable simultaneously instead of just considering bivariate joint dependency.

The application of the 3D (or any higher dimension) copula in hydro-meteorological modelling is minimal. Few previous works highlighted, for instance, the asymmetric, fully nested Archimedean copula [29,30].; the meta-elliptical Student's t copula [31]; the Plackett copula [32]; and the entropy copula [33]. All such frameworks have some statistical constraints and limitations when projected into higher dimensions. For example, the 3D symmetric Archimedean copula models the dependencies between multiple random variables by employing single-dependence parameters or generator functions and thus cannot preserve all pairwise dependencies [32,34]. Besides this, an asymmetric or fully nested Archimedean (FNA) copula can be much more reliable than a symmetric structure. FNA can individually approximate each random attribute pair through multiple parametric joint asymmetric functions [35–37]. The faithful preservation of all the lower-level dependencies among the targeted variables is still challenging based on the FNA structure. This framework is only effective and practical when two correlation structures are identical or near and lesser than the third correlation structure and are limited to a positive range [31]. Additionally, when considering more variables, the asymmetric FNA structure permits a narrow range of mutual dependencies [18]. Therefore, to alleviate all such statistical issues, the vine or pair-copula construction (PCC) approach is highly flexible and is a much more practical way of constructing any higher-dimensional joint dependence by mixing multiple 2D (bivariate) copulas in a stage-wise hierarchical nesting procedure or conditional mixing procedure [38–41].

In CE modelling, Bevacqua et al. [42] introduced a 3D vine copula for evaluating flooding events in Ravenna, Italy. In a recent study, Jane et al. [43] introduced the vine framework in the trivariate joint analysis of rainfall, ocean-side water-level and groundwater-level observations in South Florida, USA. Besides the above two, other studies—for instance, Graler et al. [41], Saghafian and Mehdikhani [44], Tosunoglu et al., [45], Latif and Mustafa [46]—often incorporated a vine copula under parametric distribution settings, thereby fitting the parametric-

class 2D copulas with parametric marginal pdfs in the parametric fitting procedure. In some previous literature, such as Silverman [47], Moon and Lall [48], Sharma et al. [49], Kim et al. [50] and Karmakar and Simonovic [51], the performance of nonparametric kernel density estimation (KDE) has been revealed to be much better than those of parametric family functions. Due to the absence of any prior distribution assumption about their marginals probability density function (PDF) type, KDE performed much more reliably, especially suited for multimodal random samples. However, the copula function eliminates the restriction to model any marginal distribution from the same family functions. The subjective assumption of the joint PDF type of the fitted parametric copulas in the traditional vine distribution framework is not much more effective at approximating joint structure, which would be questionable. In other words, fixing the joint PDF of the dependence structure to any specific or predefined copula class may fail to fully acknowledge the flexibility of the copula fitted in the vine tree structure. Parametric copulas are frequently used because of their simplicity. However, the parameter estimation procedures of the fitted parametric models are time consuming using standard statistical techniques [52]. Rauf and Zeephongsekul [53] claimed that it could lead to spurious inferences and be challenging if the underlying assumptions of the fitted parametric distribution are violated. Fitting an appropriate parametric copula demands much more attention and extra caution, which might bear the risk of uncertainty in their estimated joint exceedance values if an inappropriate dependence structure is selected.

To deal with all the above-raised issues, introducing the nonparametric copula density in the vine copula construction could be a better alternative where the 2D copulas could adapt to any dependence structure without having any specific or fixed joint PDF form. To do this, the Bernstein copula estimator and beta kernel copula density could be a good choice for modelling multivariate copula density in nonparametric settings [54–58] and reference therein. In reality, the Bernstein copula can provide higher consistency and lack boundary bias problems [59], resulting in a better estimation of the underlying dependence structure than an empirical copula estimate. Besides this, there is the performance of beta kernel density is already proved by Rauf and Zeephongsekul [53] and Latif and Mustafa [22]. The nonparametric copula density gained more attention in economics but is rarely accepted in hydro-meteorological studies. Additionally, all the above nonparametric frameworks are often limited to bivariate cases.

The main contribution of the present work is the first to incorporate the Bernstein estimator and Beta kernel copula estimator in the nonparametric estimation of the 3D vine copula density in the trivariate modelling of compound flooding (CF) events on the west coast of Canada. The objective of the present study is (i) to incorporate and test the efficacy of above-mentioned nonparametric copula densities in establishing the D-vine structure and in determining trivariate joint cumulative distribution functions (JCDF),(ii) comparing the performance with the semiparametric approach in the vine copula density, introducing parametric copulas with nonparametric marginal pdfs and the parametric approach in the vine copula. Finally, the selected best-fitted vine copula density is employed to estimate trivariate joint return periods and in assessing hydrologic risk. Our recent study is the first that incorporates the Bernstein estimator in flood modelling and confirms that this function performed well compared to Beta copula density in the bivariate dependence modelling of storm surge and rainfall events [60]. Our present study extends the previous bivariate approach by dealing with three variables, integrating the impact of river discharge events with storm surge and rainfall events in the risk of compound flooding (CF) events.

Pirani and Najafi's [61] study already shows that the higher risk of compound extreme on Canada's west coasts is due to the joint impact of precipitation, extreme water level (also, storm surge events) and streamflow discharge. Additionally, west or Pacific Canada's coast experienced higher coastal instability because of the higher risk of coastal water levels [62]. This paper is organized into four sections. After the introduction, the required theoretical background of the nonparametric copula density and in development of the 3D vine copula framework are discussed in Section 2. Section 3 of this manuscript presents the application

of the developed trivariate distribution framework to a case study in compounding the joint impact of rainfall, storm surge and river discharge events. This section comprises, for instance, modelling univariate marginal distribution via nonparametric KDE, constructing D-vine structure in the nonparametric fitting procedure via Bernstein and Beta copula estimator, D-vine structure under the parametric fitting procedure and in the semiparametric settings. This section also compares the model adequacy and performance of all three developed D-vine structures in the trivariate CF dependence. Additionally, the best-fitted trivariate structure is employed in estimating primary joint return periods for both AND and OR-joint cases and also employed in the estimation of FP statistics. Finally, Section 4 provided the research summary and conclusions.

## 2. Methodology

### 2.1. Nonparametric Copula Density Estimator

Figure 1 illustrates the methodological workflow used in this study. Firstly, compound flood variables' marginal distributions are modelled using the nonparametric kernel density estimation (KDE). The best-fitted parametric family distributions are adapted from our previous study [63], and their performance is compared with the selected KDE in the present study. The D-vine copula framework is developed under parametric, semiparametric and nonparametric settings, and their performance is compared in describing the most parsimonious flood dependence. The nonparametric vine density comprises multiple 2-D copulas via the Bernstein and Beta kernel density with kernel density margins without having any prior assumption about their marginal pdf and joint density function. The parametric and semiparametric vine copula density defines through parametric class 2-D copulas (i.e., Archimedean and Extreme value) with parametric and nonparametric via KDE margins. The best-fitted trivariate structure is employed in the estimation of trivariate joint return periods for both OR-and AND-joint cases and is further employed in estimating FP statistics. In this study, the D-vine copula are developed for three different cases, each defined by permutating the location of the conditioning variable. For instance, in case 1, the river discharge event is a conditioning variable; in case 2, the storm surge event is a conditioning variable; in case 3, the rainfall event is a conditioning variable.

Mirror image modification, transformed kernels, boundary kernels, etc., are a few examples of nonparametric approaches in joint density estimation [64–66]. This study introduces the beta kernel copula and Bernstein copula estimator for developing the D-vine structure for the trivariate joint analysis of storm surge, rainfall and river discharge events in relation to flood risk in the coastal regions. The beta kernel copula density was discussed earlier by Brown and Chen [67], Harrell and Davis [54] and Chen [68]. It is naturally free of boundary bias problems which are often encountered in the standard kernel estimator. The consistency remains in the beta kernel density if the actual density is unbounded at the boundary [69].

The 1D beta kernel density function for the given univariate variables, $A_1, A_2, \ldots, A_t$, is estimated by:

$$s_h(a) = \frac{1}{t} \sum_{i=1}^{t} K(A_i, \frac{a}{h} + 1, \frac{1-a}{h} + 1) \tag{1}$$

where "h" is the kernel's bandwidth.

In Equation (1), the density of the beta kernel function with parameters q and v is estimated by

$$K(a, q, v) = \frac{a^q (1-a)^v \Gamma(q) \Gamma(v)}{\Gamma(q+v)}, \, a \in [0,1] \tag{2}$$

According to Charpentier et al. [52], multiplying the beta kernel densities can result in beta copula joint density, known as the beta kernel copula, at point (a, b), as given below.

$$c_h(a, b) = \frac{1}{ph^2} \sum_{i=1}^{p} K(A_i, \frac{a}{h} + 1, \frac{1-a}{h} + 1) \times K\left(B_i, \frac{b}{h} + 1, \frac{1-b}{h} + 1\right) \tag{3}$$

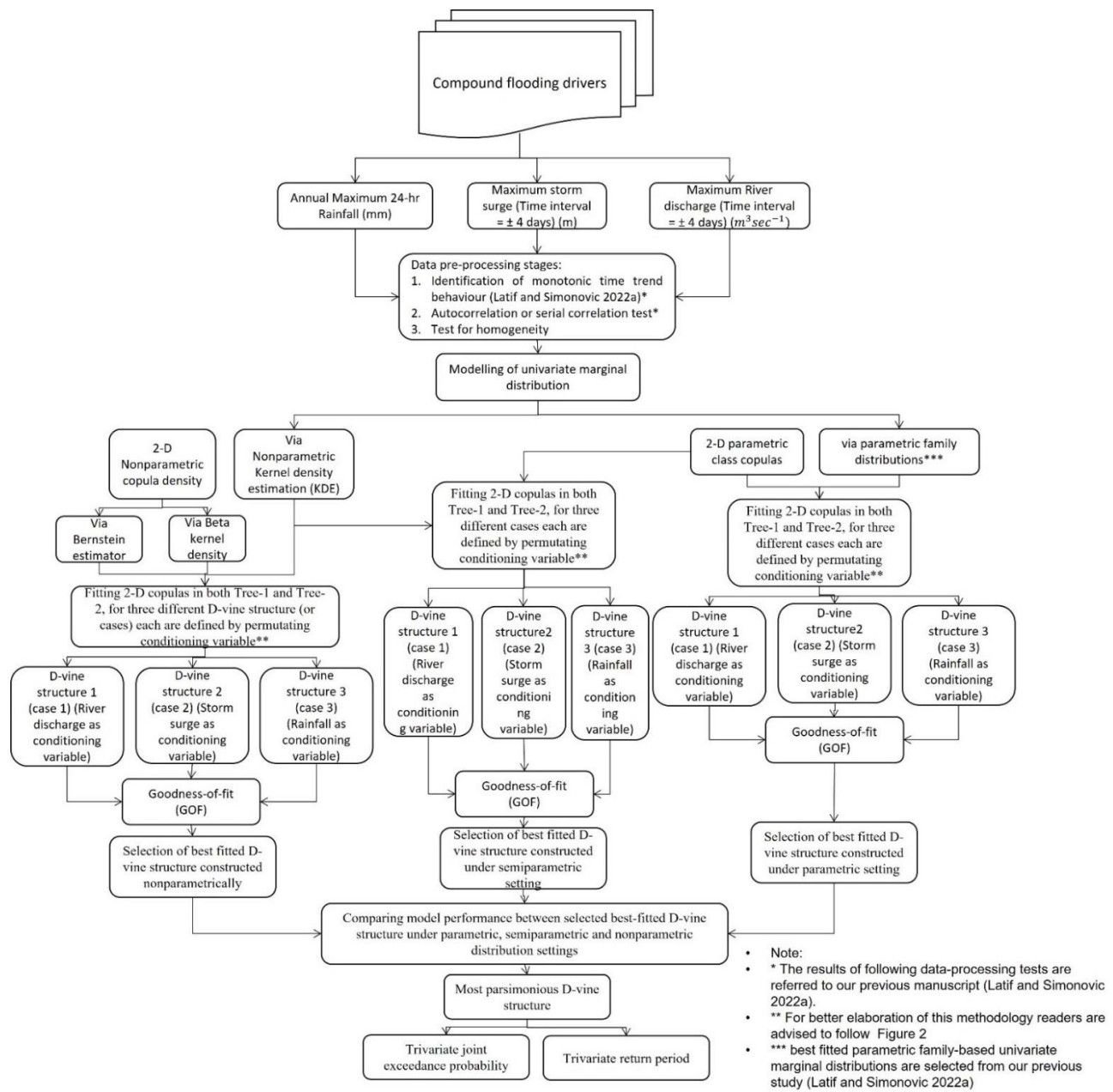

**Figure 1.** Workflow chart of the present study.

The bandwidth of Equation (3) is estimated by the rule of thumb (ROT) estimation procedure, which is based on minimizing the asymptotic mean-integrated-squared error (AMISE) statistics. For this, Nagler [70] pointed out the applicability of the Frank copula as the reference copula. The ROT bandwidth estimation for the fitted 2D beta kernel estimator of Equation (3) is estimated by

$$h = \left( \frac{1}{8\pi} \frac{\varsigma(c)}{\xi(c)} \right)^{\frac{1}{3}} n^{\frac{-1}{3}} \tag{4}$$

where "c" is assumed to be the Frank copula in Equation (4).

The efficacy of the Bernstein copula estimator is also tested and compared in constructing the D-vine structure together with beta kernel density. Lorentz [71] highlighted that the Bernstein polynomial could be used to approximate any continuous functions within a range of [0,1]. Tenbush [72] constructed bivariate joint density using the Bernstein

estimator. Approximation of nonparametric joint density using the Bernstein copula can provide higher consistency and remove boundary bias problems [57,73]. Additionally, it can better estimate the underlying mutual correlation and good approximation with an asymmetric and extreme dependency compared to an empirical copula approach [69].

Mathematically, the n-degree Bernstein polynomial is estimated by [57,69]:

$$B(n, w, z) = \binom{n}{w} z^w (1 - z)^{n-w} \tag{5}$$

In Equation (5), $w = 0, 1, 2, \ldots, n \in \mathbb{N}; 0 \leq z \leq 1$.

Now, if $X = (X_1, X_2)$ illustrates bivariate observations having a uniform marginal distribution over $Y_i = \{0, 1, 2, \ldots, n_i\}$ with grid size $n_i \in \mathbb{N}$ and where $i = 1, 2$, then

$$y(w_1, w_2) = Y\left(\cap_{I=1}^2 \{X_i = w_i\}\right), (w_1, w_2) \in [0, 1]^2 \tag{6}$$

Hence, for the 2D joint distribution case, the Bernstein copula density is estimated by;

$$c(x_1, x_2) = \sum_{w_1=0}^{n_1-1} \sum_{w_2=0}^{n_2-1} y(w_1, w_2) \prod_{i=1}^2 n_i B(n_i - 1, w_i, x_i), \tag{7}$$

where $(x_1, x_2) \in [0, 1]^2$.

### 2.2. Univariate Kernel Density Estimation of Flood Margins

Parametric class functions are often restricted to prior distributional assumptions about their univariate marginal pdfs. However, on the other side, the parametric functions performed well if the given observation exhibited unimodality or symmetrical behaviour. A nonparametric kernel density estimation (KDE) is identified as much more robust and better performing in modelling the probability densities of different hydro-meteorological characteristics, especially when the given observation departed from the symmetrical behaviour or, say, bi- or multimodality [48,50,74,75]. Our present study tested the efficacy of different KDE functions and compared their performance with the selected best-fitted parametric distributions from our previous study [63].

Mathematically, the 1D kernel function can approximate a probability density structure having the following statistical property.

$$\int_{-\infty}^{+\infty} K(x) dx = 1 \tag{8}$$

Furthermore, the kernel function can be represented by a general function:

$$K_o(x) = \frac{1}{o} K\left(\frac{x}{o}\right) \tag{9}$$

where "o" is the bandwidth of the fitted univariate kernel function.

By taking the average of Equation (9), the univariate kernel density estimator $\hat{f}_o(x)$ of the probability density function is estimated by

$$\hat{f}_o(x) = \frac{1}{po} \sum_{i=1}^p K_o\left(\frac{x - X_i}{o}\right) \tag{10}$$

where "p" is the observation counts. In fitting the kernel density to the given observational samples, selecting an appropriate way of estimating kernel bandwidth is often essential; otherwise, it may be attributed to either over-smoothing or under- or insufficient smoothing (also called rough smoothing). For extended details about different statistical procedures in kernel bandwidth estimation, readers are advised to read Sharma et al. [49] and Jones et al. [76]. In our present analysis, the direct plug-in (DPI) method is used to esti-

mate the bandwidth of the fitted kernel density [77–79]. Table 1 lists some kernel density functions which are used in this study.

**Table 1.** 1-D Kernel density estimation (KDE) tested in the modelling of flood marginals.

| SI No. | Kernel Density Estimation | K(x) |
|:---:|:---:|:---:|
| 1 | Normal | $= (2\pi)^{\frac{-1}{2}} e^{-(x^2)/2}$ |
| 2 | Epanechnikov (or parabolic) | $= 0.75(1 - x^2),\ \|X\| \leq 1$ <br> $= 0,\ \text{otherwise}$ |
| 3 | Biweight (or Quartic) | $= 0.9375\left(1 - x^2\right)^2,\ \|x\| \leq 1$ <br> $= 0,\ \text{otherwise}$ |
| 4 | Triweight | $1.09375\left(1 - x^2\right)^3,\ \|x\| \leq 1$ <br> $= 0,\ \text{otherwise}$ |

### 2.3. D-Vine Copula Structure in the Trivariate Modelling

The vine or pair-copula construction (pcc) is based on the decomposition of full multivariate density into a cascade of local blocks of the best-fitted 2D copulas modelled between each random pair and their conditional and unconditional functions [38,39]. Two famous decomposition steps in the vine framework are the canonical or C-vine and drawable or D-vine [80,81]. The D-vine's structure is highly flexible, and this is accepted widely [40,41,82]. The traditional approach in the vine framework considers multiple 2D parametric class copulae in the stagewise hierarchy. A few statistical constraints with parametric copula joint density are highlighted in Section 1. Therefore, it could be problematic if the vine copula is constructed using the parametric class copulas. Due to this, we individually tested the efficacy of the nonparametric method via beta kernel copula density and Bernstein copula estimator in the 3D vine simulation for the given CF variables. This study also compares the performances of the parametric and semiparametric approaches in the vine simulation, where both frameworks are defined through multiple 2D parametric copulas. The univariate marginal distribution is modelled using the kernel density estimations (KDE) in both nonparametric and semiparametric and parametric class distributions in the parametric vine approach.

Due to the involvement of three flood characteristics in characterizing C.F. events in our study, the present 3D vine framework must demand three 2D copulae and two tree levels, Tree 1 and Tree 2 (refer to Figure 2). For trivariate variables (A, B, C), the D-vine structure can be mathematically expressed as

$$f(a, b, c) = f(a) \cdot f(b|a) \cdot f(c|a, b) \tag{11}$$

$$f(b|a) = \frac{f(a, b)}{f(a)} = c_{ab}(F(a),\ F(b)) \cdot f(b) \tag{12}$$

$$f(c|a, b) = \frac{f(b, c|a)}{f(b|a)} = c_{bc|a}(F(b|a),\ F(c|a)) \cdot c_{ac}(F(a),\ F(c)) \cdot f(c) \tag{13}$$

In Equation (11), the conditional cumulative distribution functions $f(b|a)$ and $f(c|a, b)$ are estimated using the pair-copula densities. Additionally, F(a), F(b) and F(c) are the fitted univariate margins. In Equation (12), $C_{ab}$ is the best-fitted 2D copula (parametric class or nonparametric) for variables A and B. Our proposed framework selects the D-vine with five nodes, three edges and two tree levels (Refer to Figure 2). We constructed a D-vine copula framework for three cases. Each case was defined based on permutating the conditioning variables (or variable placed at the centre of the selected D-vine structure; refer to Figure 2). For instance, the D-vine structure 1 (case-1) was defined by selecting the river discharge (RD) as a conditioning variable placed between storm surge (SS) and rainfall (R) events. Similarly, D-vine structure 2 (case 2) and D-vine structure 3 (case 3) are defined by placing storm surge and rainfall events as conditioning variables (refer to Figure 2). This

permutation approach to considering each variable of interest as a conditioning variable and selecting the best-fitted vine structure using the fitness test statistics can provide a much more practical and flexible way to the vine copula approach.

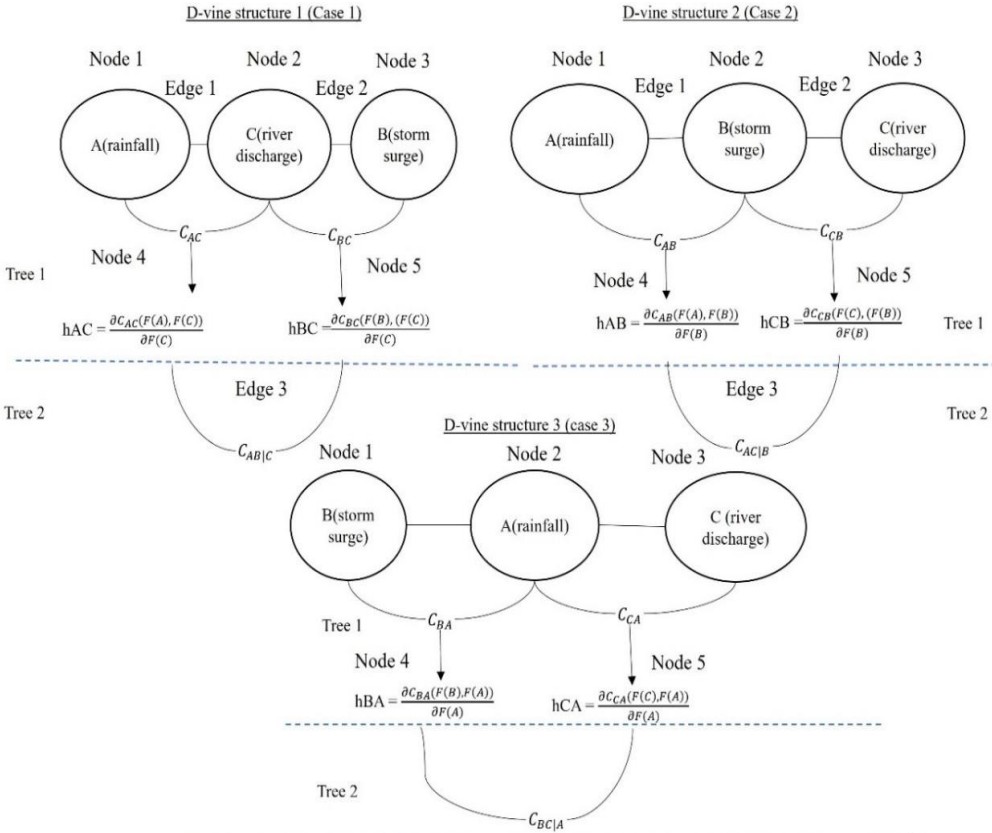

- F(A) = best fitted univariate marginal distribution (using parametric or univariate functions) of annual maximum 24-hr rainfall (mm)
- F(B) = best fitted univariate marginal distribution (using parametric or nonparametric functions) of maximum storm surge (Time interval = ± 4 days)
- F(C) = best fitted univariate marginal distribution (using parametric or nonparametric functions) of maximum river discharge (Time interval = ± 4 days)
- $C_{AC}$ (best fitted 2-D copula between flood pair A and C; $C_{BC}$ (between B and C); $C_{AB}$ (between A and B).
- hAC = conditional cumulative distribution functions (CCDF) estimated using the partial derivatives of $C_{AC}$ w.r.t F(C); hAB = CCDF using partial derivative of $C_{AC}$ w.r.t to F(B); hBC = CCDF using partial derivative of $C_{BC}$ w.r.t to F(C); hBC = CCDF using partial derivative of $C_{BC}$ w.r.t to F(B); hBA = CCDF using partial derivative of $C_{BA}$ w.r.t to F(A); hCA = CCDF using partial derivative of $C_{CA}$ using partial derivative of F(A);

**Figure 2.** Schematic diagram in the 3-D D-vine copula simulation for three different cases [Note: each case of the D-vine structure is defined by permutating the location of the conditional variable, for instance, D-vine structure-1 (River discharge as conditioning variable), D-vine structure-2 (Storm surge as conditioning variable), D-vine structure-3 (Rainfall events as conditioning variable)].

Refer to Figure 2 (consider either case) for illustration; the best-fitted univariate flood marginal distribution is selected after selecting the conditioning or centred variable (say B or A or C). In constructing the D-vine framework nonparametrically, kernel density estimation (KDE), obtained from Section 2.2, is selected to define flood marginal probability distribution. After that, nonparametric copula density (refer to Section 2.1) is introduced and tested via the Bernstein estimator and the beta kernel density estimator. Thus best-fitted models are selected using the fitness test statistics for different tree levels (Tree 1 and Tree 2, refer to Figure 2).

At first, using the most parsimonious 2D copulas, either parametric class (refer to Latif and Simonovic [63]) or nonparametric (refer to Section 2.1), are selected for each flood pair, say $C_{AB}$ and $C_{C.B.}$, the conditional cumulative distribution function (CCDFs), also called the h-function, is estimated [41,82].

$$F_{A|B}(a, b) = h_{A.B.} = \frac{\partial C_{A\,B}(F(A),\ F(B))}{\partial F(B)} \text{ and } F_{C|B}(C, B) = h_{C.B.} = \frac{\partial C_{C\,B}(F(C),\ F(B))}{\partial F(B)} \tag{14}$$

In the second Tree 2 level (refer to Figure 2), the CCDFs statistics estimated from Tree 1 level, using copula $C_{AB}$ and $C_{C.B.}$, is now input to describe another 2D copula in the modelling of joint dependence of conditional pair (AC | B), such as $C_{AC|B}$.

In the nonparametric vine copula approach, the 2D Bernstein copula estimator (refer to Equation (7)) and beta kernel copula density (refer to Equation (3)) are tested individually in both tree levels in the D-vine structures (for all three cases, refer to Figure 2). Our recent study tested the adequacy of different parametric copulas, for instance, mono-parametric Archimedean copulas, mixed or bi-parametric Archimedean copulas and rotated versions (by 180 degrees) of mixed Archimedean copulas, etc., for the same flood pairs [63]. The selected best-fitted 2D copulas from our previous study are now employed in fitting bivariate flood pairs and estimating CCDFs in the first tree level (Tree 1), of the parametric and semiparametric-based vine framework.

Finally, after selecting the most justifiable copula for each tree level for each D-vine structure (case 1, case 2 and case 3) finally, the full trivariate joint density is calculated by

$$C_{A\,B\,C}(a,b,c) = C_{A\,C|B}\Big(F_{A\backslash B}(a,b),\ F_{C|B}(c,b)\Big)\ \cdot C_{A\,B}\cdot C_{C\,B} \tag{15}$$

*2.4. Trivariate Joint Return Periods*

Frequency analysis provides a mathematical relationship between extreme events quantiles and their non-exceedance probabilities (or return period) by fitting the most justifiable univariate or multivariate probability distribution function [83,84]. The return period measures the mean or average inter-arrival time between the two design events [85]. The univariate return period's validity is questionable in multidimensional extremes like compound flooding due to the joint action of multiple drivers. In our current study, the developed 3D joint framework is applied to estimate primary return periods, which are further defined in two cases: OR-joint return period and AND-joint return period [86–89]. Different notations of return periods have their own importance that could depend upon the nature of the undertaken problem. For example, just considering an OR-joint return period or either AND-joint return period would be problematic [31]. A practical risk assessment approach must consider different approaches in the return period estimations; readers are advised to see Graler et al. [41] and Requena et al. [90].

Consider the trivariate events (A ≥ a OR B ≥ b OR C ≥ c), where either of the events exceeds a specific threshold value; the OR-joint return periods are estimated using the trivariate joint exceedance probability given below.

$$T^{OR}_{A,\,B,C}(a,b,c) = \frac{1}{P\,(A \geq a\ \lor\ B \geq b\ \lor C \geq c)} = \frac{1}{(1 - C(F(a),\ F(b),\ F(c)))} \tag{16}$$

where $C(F(a),\ F(b),\ F(c))$ is the trivariate joint cumulative distribution function (JCDF) estimated using the best-fitted 3D vine copula structure.

Similarly, consider another trivariate joint dependency case (A ≥ a AND B ≥ b AND C ≥ c) where all the events exceed a specific threshold value simultaneously; the AND-joint return periods are estimated by considering both trivariate joint cumulative distribution function (JCDF) and bivariate JCDFs which are defined for each random flood pair given below.

$$T^{AND}_{A,\,B,\,C}(a,b,c) = \frac{1}{P\,(A \geq a\ \land\ B \geq b\,\land\ C \geq c)} = \frac{1}{(1 - F(a) - F(b) - F(c) + C(F(a),\ F(b)) + C(F(b),F(c)) + C(F(a),F(c)) - C(F(a),F(b),F(c)))} \tag{17}$$

In Equation (17), $C(F(a),\ F(b))$, $C(F(b),\ F(c))$ *and* $C(F(a),\ F(c))$ are the bivariates (JCDFs) obtained by fitting most parsimonious 2D copulas to targeted random pairs, and $C(F(a), F(b), F(c))$ is the JCDF using the fitted 3D copula density.

*2.5. Failure Probability in the Hydrologic Risk Evaluation*

In the hydrologic risk assessments of CE, consideration of only traditional joint primary return periods would be ineffective in describing the risk of potential flood events during the entire project design lifetime [10,91]. In recent studies, a hydrologic risk tool called failure probability (FP) statistics [92,93] is highlighted and used efficiently. FP usually defines the chance of potential flood hazards occurring at least once in a given project design lifetime. FP statistics can define the risk of CF events more appropriately than just visualizing their joint return periods. Few studies incorporated FP statistics in the bivariate hydrologic risk assessments [10,94]. This study incorporated FP statistics in the trivariate compound flood risk assessment, which can be mathematically expressed as

$$FP_T = 1 - (1 - P)^T \tag{18}$$

where T is the arbitrary project lifetime.

Similarly, for the trivariate flood hazard scenario, the risk of failure for the OR-joint case can be estimated by;

$$FP_T = 1 - (1 - P\,(\text{Rainfall} \geq r\ \text{OR Storm surge} \geq s\ \text{OR River discharge} \geq rd)^T \tag{19}$$

## 3. Application

### 3.1. Study Area and Defining the Compound Hazard Scenario

The complex interplay between oceanographic, fluvial and pluvial factors increases the risk of extreme devastation in low-lying coastal communities worldwide. This study introduces a nonparametric approach to constructing a 3D vine copula framework in compounding the collective impact of rainfall, storm surge and river discharge in flooding events. Our work introduces 46 years of selected flood characteristics collected at west Canada's coast in the trivariate joint probability analysis. The low-lying regions near the Pacific coast and Fraser River are highly susceptible to flooding and often encounter mature and extra-large tropical storms. When these storms are encountered in the coastal mountains, they can result in devastating disasters, forming the potential for prolonged impact. Fraser River is the longest river in the south of Metro Vancouver, BC, with an annual discharge at its river mouth of 3550 m$^3$s$^{-1}$. This river flows for 1375 km and finally drains out into the Strait of Georgia. Pirani and Najafi's [61] study already identified that the joint combination of tidal water extreme level, precipitation and river discharge can increase the risk of coastal flooding at the Pacific west coast of Canada. The risk of extreme water levels increases the risk of storm surge events. The same scenario can result in devastating hydrologic or compound flooding when combined with high river discharge and extreme rainfall events. The Environment Ministry of BC report [95] also reported the expectation of a rise in sea level by about half a meter by the end of this 2050 and one meter by the end of 2100. Besides this, according to Lemmen et al. [96], the impact of climate change across Canada significantly increases the risk of extreme events.

This study searches the dependency for the annual maximum 24 h rainfall events and their associated river discharge and storm surge events observed within a time lag of ±4 days from the date of annual maximum 24 h rainfall events. Our previous study [63] has already confirmed that more significant dependencies can be observed when considering the maximum storm surge and river discharge events within a time lag of ±4 days from the calendar date of the annual maximum 24 h Rainfall events. At first, the coastal water level (CWL) data were obtained of 1970 to 2018 from the New Westminster tidal gauge station (station id = 7654) with their geographical coordinates (49.2° N Lat and 122.9° W Lon), which were delivered by Fisheries and Ocean Canada. Secondly, the storm-surge data were estimated by differencing observed CWL data and predicted water level or astronomical tide data, which requires proper time matching between them. Canadian Hydrographic Services (CHS) delivered the predicted tide data. Similarly, the rainfall data were collected at by Haney UBC RF Admin gauge station (49°15′52.1″ N Lat and 122°34′400″ W Lon).

Both storm-surge data and rainfall data were collected for the same calendar year. Third, Environment and Climate Change Canada provided the streamflow discharge data collected at the Fraser River at Hope (49°23′09″ N Lat and 121°27′15″ W Lon). It should be noted that the nearest rainfall gauge station and streamflow discharge station were selected within a radial distance of 50 km centring the selected tidal gauge station.

The annual maximum 24 h rainfall data were defined for each year using the daily-basis rainfall events. The river discharge and storm surge data were selected by observing their maximum values within a time lag of ±4 days from the date of annual maximum 24 h rainfall events. Due to the missing data in the period between the years 1970 to 2018, we considered 46 years of data in establishing a trivariate compound relationship between the variable of interest. Supplementary Table S1 lists the descriptive statistics of targeted compound flood (CF) driving agents. Supplementary Figure S1, S2a–c and S3a–c illustrate the box plots, histogram plots and normal quantile-quantile (Q-Q) plots. From Figure S3c it was found that river discharge observations exhibited more deviation from normality (or straight line) compared to storm surge (Figure S3b) and rainfall events (Figure S3a).

*3.2. Nonparametric Estimation in the Univariate Flood Marginals*

Modelling the univariate flood marginal is a statistical procedure to infer the population based on a finite random sample and is often a mandatory pre-requisite. Our previous study, using the same dataset, confirmed that annual maximum 24 h rainfall and maximum river discharge events exhibit no serial correlation and zero monotonic trends within their time series [63]. Conversely, maximum storm surge (Time interval = ±4 days) events have zero serial correlation but exhibit monotonic time trend behaviour, which is estimated using the nonparametric Mann–Kendall (M-K) test [97,98] at 5% significance (or 95% confidence interval). Besides this, homogeneity tests for the given time series were examined to identify if changes occur within time series of flood characteristics, using Pettitt's test [99], the SNHT (standard normal homogeneity test) [100] and Buishand's test [101]; refer to Supplementary Table S2. It was found that both rainfall and river discharge events exhibited homogenous behaviour, but storm surge events showed non-homogenous characteristics. In the second row of Table S2, the estimated *p*-value for storm surge events is less than 0.05 for both the Pettit and SNHT tests. In conclusion, an independent and identical distribution (i.i.d.) is required before introducing it into the probability distribution framework. Thus, a differencing procedure was adopted to remove non-stationarity or de-trend storm surge observations [63].

Table 1 introduces some frequently used kernel density estimations (KDE), whose efficacy was tested in this study to model univariate flood margins. The bandwidth of the fitted KDE was estimated using the direct plug-in (DPI) algorithm; refer to Section 2.2. Table 2a–c list the fitted KDE and their estimated bandwidth. The adequacy of the fitted non-parametric KDE models was tested by comparing empirical and theoretical probabilities. The empirical probabilities were estimated using the Gringorten-based position-plotting approach for each flood characteristic [102]. The cumulative distribution function (CDF) of the fitted KDE was estimated via numerical integration technique or empirical approach because of the lack of a closed form of probability density and cumulative distribution [50]. The goodness-of-fit (GOF) tests, such as mean-square error (MSE), root mean-square error (RMSE), Akaike information criterion (AIC), Bayesian information criterion (B.I.C.), Hannan–Quinn information criterion (H.Q.C.) and mean absolute error (MAE), were estimated for each fitted model [103–107] refer to Table 2a–c. It was found that normal KDE performed best (minimum value of MSE., RMSE, AIC, BIC, HQC and MAE statistics) and was selected for defining the marginal probability density function (PDF) of the maximum 24 h rainfall, maximum storm surge (Time interval = ±4 days) and maximum storm surge (Time interval = ±4 days) events. The qualitative or graphical investigation, using the comparative C.D.F. plots and probability–probability (P-P) plots (refer to Supplementary Figure S4a–c and S5a–c), confirmed the suitability of the selected normal KDE.

**Table 2.** Fitting univariate kernel density estimation (KDE) and their goodness-of-fit (GOF) test for (a) Annual maximum 24 h Rainfall (mm) (b) Maximum Storm surge (Time interval = ±4 days) (m) (c) Maximum River discharge (Time interval = ±4 days) ($m^3s^{-1}$).

**(a)**

| Nonparametric KDE | Estimated Bandwidth (via Direct-Plug-in Method) | MSE (Mean Square Error) | RMSE (Root Mean Square Error) | AIC (Akaike Information Criterion) | BIC (Bayesian Information Criterion) | HQC (Hannan-Quinn Information Criterion) | MAE (Mean Absolute Error) |
|---|---|---|---|---|---|---|---|
| Normal * | 11.25 | 0.0003 | 0.0199 | −358.22 | −356.39 | −357.53 | 0.015 |
| Epanechnikov (or parabolic) | 24.90 | 0.0007 | 0.0281 | −326.36 | −324.53 | −325.67 | 0.023 |
| Biweight (or Quartic) | 29.50 | 0.0010 | 0.0316 | −315.56 | −313.73 | −314.88 | 0.026 |
| Triweight | 33.50 | 0.0011 | 0.0342 | −308.42 | −306.59 | −307.73 | 0.027 |
| Parametric GEV (Latif and Simonovic 2022a [63]) ** | Estimated parameters via Maximum likelihood estimation (MLE) <br><br> location(mu = $\mu$) = 1494.64; scale (sigma = $\sigma$) = 616.37; shape (xi = $\xi$) = 0.31 | 0.0009 | 0.0312 | −312.97 | −307.48 | −310.91 | 0.024 |

**(b)**

| Nonparametric KDE | Estimated Bandwidth (via Direct-Plug-in Method) | MSE (Mean Square Error) | RMSE (Root Mean Square Error) | AIC (Akaike Information Criterion) | BIC (Bayesian Information Criterion) | HQC (Hannan-Quinn Information Criterion) | MAE (Mean Absolute Error) |
|---|---|---|---|---|---|---|---|
| **Normal *** | **0.07** | **0.0003** | **0.0175** | **−369.83** | **−368.00** | **−369.15** | **0.014** |
| Epanechnikov (or parabolic) | 0.16 | 0.0007 | 0.0265 | −331.95 | −330.12 | −331.26 | 0.020 |
| Biweight (or Quartic) | 0.20 | 0.0008 | 0.0288 | −324.17 | −322.34 | −323.48 | 0.022 |
| Triweight | 0.22 | 0.0009 | 0.0304 | −319.30 | −317.47 | −318.61 | 0.023 |
| Parametric Normal (Shahid and Simonovic 2022a [63]) ** | Estimated parameters via Maximum likelihood estimation (MLE) <br><br> mean (mu = $\mu$) = 2.340757e−18; sd (sigma = $\sigma$) = 1.676386e−01 | 0.0011 | 0.034 | −306.56 | −302.90 | −305.19 | 0.026 |

**Table 2.** *Cont.*

(c)

| Nonparametric KDE | Estimated Bandwidth (via Direct-Plug-in Method) | MSE (Mean Square Error) | RMSE (Root Mean Square Error) | AIC (Akaike Information Criterion) | BIC (Bayesian Information Criterion) | HQC (Hannan-Quinn Information Criterion) | MAE (Mean Absolute Error) |
|---|---|---|---|---|---|---|---|
| **Normal *** | **340.21** | **0.0005** | **0.0223** | **−347.61** | **−345.78** | **−346.93** | **0.017** |
| Epanechnikov (or parabolic) | 753.16 | 0.0017 | 0.0415 | −290.65 | −288.82 | −289.96 | 0.030 |
| Biweight (or Quartic) | 892.25 | 0.0019 | 0.0444 | −284.42 | −282.60 | −283.74 | 0.033 |
| Triweight | 1013.19 | 0.0022 | 0.0477 | −277.81 | −275.99 | −277.13 | 0.037 |
| Parametric GEV (Shahid and Simonovic 2022a [63]) ** | Estimated parameters via Maximum likelihood estimation (MLE) <hr> location (mu = $\mu$) = 1494.64; scale (sigma = $\sigma$) 616.37; shape (xi = $\xi$) = 0.31 | 0.0008 | 0.0291 | −319.15 | −313.67 | −317.10 | 0.022 |

(a) Note: Normal KDE (bold letter with single asterisk *) outperformed (minimum value of MSE, RMSE, AIC, BIC, HQC and MAE), thus selected in defining the univariate marginal distribution of Annual maximum 24 h Rainfall (mm) events. Additionally, the GEV distribution (double asterisk **) was selected as best-fitted when comparing the performance of different 1-D parametric family distributions in modelling Annual maximum 24 h Rainfall (mm) events (Latif and Simonovic 2022a [63]). (b) Note: Normal KDE (bold letter with single asterisk *) outperformed (minimum value of MSE, RMSE, AIC, BIC, HQC and MAE), thus selected in defining the univariate marginal distribution of Maximum Storm surge (Time interval = ±4 days). Additionally, Normal distribution (double asterisk **) selected as best-fitted when comparing the performance of different 1-D parametric family distributions in modelling storm surge events (Latif and Simonovic 2022a [63]). (c) Note: Normal KDE (bold letter with single asterisk *) outperformed (minimum value of MSE, RMSE, AIC, BIC, HQC and MAE), thus selected in defining the univariate marginal distribution of Maximum River discharge (Time interval = ±4 days). Additionally, GEV distribution (double asterisk **) was best fitted when comparing the performance of different 1-D parametric family distributions in modelling river discharge events (Latif and Simonovic 2022a [63]).

Our previous study selected the generalized extreme value (GEV), normal and GEV distribution fit that were best for the same dataset tested in the present study [63]). The nonparametric KDE outperformed the others (refer to Table 2).

### 3.3. Incorporation of Nonparametric Vine Structure in the Trivariate Flood Dependence

Our previous study [63] already confirmed the existence of positive dependence between flood attribute pairs, which was measured both parametrically via Pearson correlation coefficient and nonparametric via Kendall's tau ($\tau$), and Spearman's rho ($\rho$) at a 5% significance level (95% confidence interval). At first, the nonparametric via 2D Bernstein estimator and beta kernel estimator (refer to Equations (7) and (3)) were employed in the bivariate dependence modelling of the rainfall–storm-surge, storm surge–river-discharge and rainfall–river-discharge pairs (refer to Table 3. The beta kernel density and Bernstein copula estimator can alleviate the risk of boundary bias problems. The Bernstein copula can facilitate higher consistency and better approximate joint structure than the empirical copula. The fitted beta kernel density bandwidth was examined using the rule of thumb (ROT) approach by minimizing the AMISE statistics (refer to Equation (4) of Section 2.1). Similarly, in fitting the 2D Bernstein copula estimator, their coefficient was adjusted by the approach discussed by Weiss and Scheffer [58].

**Table 3.** Fitting the nonparametric 2-D copula density and their goodness of fit (GOF) test to the given flood attribute pair.

| Flood Attribute Pairs | Nonparametric 2-D Copula Density | Estimated Bandwidth (Only for Beta Kernel Copula Density) | MSE (Mean Square Error) | RMSE (Root Mean Square Error) | MAE (Mean Absolute Error) | K-S (Kolmogorov–Smirnov) | NSE (Nash–Sutcliffe model Efficiency Coefficient) |
|---|---|---|---|---|---|---|---|
| Annual Maximum 24 h Rainfall (mm)-Maximum Storm surge (Time interval = ±4 days) (m) | **Bernstein estimator \*** | 0.11 | **0.0013** | **0.0360** | **0.0290** | **D = 0.0869, *p*-value = 0.99** | **0.980** |
| | Beta kernel density | | 0.0014 | 0.0371 | 0.0295 | D = 0.1521, *p*-value = 0.66 | 0.979 |
| Maximum Storm surge (Time interval = ±4 days) (m)-Maximum River discharge (Time interval = ±4 days) (m³/s) | **Bernstein estimator \*** | 0.12 | **0.0012** | **0.0350** | **0.0270** | **D = 0.1521, *p*-value = 0.66** | **0.981** |
| | Beta kernel density | | 0.0014 | 0.0375 | 0.0295 | D = 0.1521, *p*-value = 0.66 | 0.978 |
| Annual Maximum 24 h Rainfall (mm)-Maximum River discharge (Time interval = ±4 days) (m³/s) | Bernstein estimator | 0.17 | 0.0011 | 0.0338 | 0.0258 | D = 0.1956, *p*-value = 0.34 | 0.981 |
| | **Beta kernel density \*** | | **0.0008** | **0.0298** | **0.0221** | **D = 0.1521, *p*-value = 0.66** | **0.985** |

Note: Bernstein copula estimator (bold letter with an asterisk) fitted best for flood pairs rainfall and storm surge, and storm surge and river discharge. Beta kernel copula density is most parsimonious for flood pair rainfall and river discharge.

The nonparametric models' performances were evaluated using various G.O.F. measures, for instance, MSE, RMSE, MAE, KS (Kolmogorov–Smirnov) [108] and NSE (Nash–Sutcliffe model efficiency coefficient) [109]; refer to Table 3. The Bernstein copula estimator's performance was better for flood pairs (rainfall and storm surge) and (storm surge and river discharge) (minimum values of MSE, RMSE, MAE and KS test and high NSE test statistic). However, according to Table 3, the beta kernel density outperformed Bernstein estimator for the rainfall and river discharge pair.

We constructed the D-vine copula for three cases. Each case defines a D-vine structure by permutating the locations of conditioning variables. All the computation involved in the establishment of 3D vine copula (also in fitting 2D nonparametric copula density) was carried out using R software [110] with the libraries "kdecopula" [111] and "kdevine" [70].

1.  D-Vine structure 1 (case 1) considers river discharge observation as a conditioning variable by placing it at the centre of the vine structure (refer to Figure 2 and Tables 3 and 4). In this structure, at first, the 2D beta kernel copula density and 2D Bernstein copula estimator, which were selected as best-fitted from Table 3 for flood pair rainfall–river-discharge and storm surge–river-discharge in the first tree level (Tree 1), were now employed in the estimation of conditional cumulative distribution functions (CCDFs); $h_{\text{RAIN RIVER DISCHARGE}}$ and $h_{\text{STORM SURGE RIVER DISCHARGE}}$ (refer to Equation (14)). The copula in the second tree level (Tree 2) was then identified using the above estimated CCDDFs values as input. It was found that the Bernstein copula estimator outperformed the beta kernel density to model joint dependence for the flood pair (RAIN, STORM SURGE | RIVER DISCHARGE) $C_{\text{RAIN STORM SURGE|RIVER DISCHARGE}}$ (which exhibited the minimum value of MSE, RMSE, MAE and KS and the higher value of NSE test statistics). Finally, the full 3D trivariate joint density was obtained using Equation (15).

2.  Similarly, D-vine structure 2 (case 2) comprises storm surge events as a conditioning variable (refer to Tables 3 and 4 and Figure 2). In this vine framework, at first, in the first tree level (Tree 1), the Bernstein estimator is identified as the most

justifiable and thus is employed in the estimation of CCDFs $h_{\text{RAIN, STORM SURGE}}$ and $h_{\text{RIVER DISCHARGE, STORM SURGE}}$, followed by Equation (14). Secondly, in the second tree level (Tree 2), the Bernstein copula estimator is selected as the most parsimonious in establishing the dependence between of flood pair (RAIN RIVER DSICHARGE | STORM SURGE) $C_{\text{RAIN RIVER DSICHARGE|STORM SURGE}}$. Finally, using Equation (15), the full trivariate joint density of the fitted vine structure is estimated.

3. D-Vine structure 3 (case 3) is defined by considering rainfall events as a conditioning variable placed in the centre of the selected D-vine structure (refer to Figure 2, and Tables 3 and 4). The Bernstein estimator and beta kernel density were identified as most justifiable and thus employed in the estimation the CCDFs $h_{\text{STORM SURGE, RAINFALL}}$ and $h_{\text{RIVER DISCHARGE, RAIN}}$ (followed by Equation (14)) in Tree 1. In the second level (Tree 2), again the Bernstein copula estimator was identified as the most parsimonious in modelling joint dependence of flood pair (STORM SURGE RIVER DSICHARGE | RAINFALL) (refer to Table 4). Finally, followed by Equation (15), the full trivariate joint density of the fitted vine structure was defined.

After approximating three different D-vine structures (case 1, case 2 and case 3), their performances were compared using the fitness test statistics (MSE, RMSE, MAE, NSE and K-S). The theoretical probability (CDF) was estimated using a developed 3D vine structure for each case and compared with empirical observations for estimating the GOF test statistics. Table 4 shows that the D-vine structure for case-2, considering storm surge as a conditioning variable, performed better than other D-vine structures. The selected structure exhibited the minimum MSE, RMSE and MAE values and a high NSE test value. The above approach in the vine copula provided much better flexibility in selecting the best vine model, not just by fixing the conditioning variable but by switching or permutating the conditioning variable. For example, in the above case, when considering storm surges as conditioning variables, the performance of the fitted D-vine copula got better than considering either rainfall or river discharge events. Supplementary Figure S8 illustrates the vine tree plot of the developed D-vine structure in the nonparametric fitting procedure.

**Table 4.** Summary statistics in the nonparametric D-vine structure for three different cases by incorporating the Bernstein estimator and Beta kernel density (each D-vine structure is defined by permutating the location of the conditioning variable).

| Vine Structure (Conditioning Variable) | Tree Level | Flood Attribute Pairs | Fitted Non-parametric Copula Estimator | Estimated Bandwidth (for Beta Kernel Copula Density) | MSE (Mean Square Error) | RMSE (Root Mean Square Error) | MAE (Mean Absolute Error) | K-S (Kolmogorov–Smirnov) | NSE (Nash–Sutcliffe Model Efficiency Coefficient) |
|---|---|---|---|---|---|---|---|---|---|
| Case 1 (1-3-2) | Tree 1 | 1-3 (Rain-River discharge) | Beta kernel | 0.11 | 0.0004 | 0.0223 | 0.017 | D = 0.086, *p*-value = 0.99 | 0.992 |
| | | 2-3 (storm surge-river discharge) | Bernstein | | | | | | |
| | Tree 2 | 12\|3 | Bernstein | | 0.0006 | 0.0249 | 0.019 | D = 0.087, *p*-value = 0.99 | 0.990 |
| | | | Beta kernel | | | | | | |
| **Case 2 (1-2-3) \*** | Tree 1 | 1-2 (Rain–Storm Surge) | Bernstein | 0.17 | **0.0002** | **0.0153** | **0.013** | **D = 0.152, *p*-value = 0.66** | **0.995** |
| | | 3-2 (Storm surge–river discharge) | Bernstein | | | | | | |
| | Tree 2 | 13\|2 | **Bernstein** | | 0.0003 | 0.0154 | 0.013 | D = 0.152, *p*-value = 0.66 | 0.995 |
| | | | Beta kernel | | | | | | |
| Case 3 (2-1-3) | Tree 1 | 2-1 (Storm surge-Rain) | Bernstein | 0.12 | 0.0004 | 0.0203 | 0.015 | D = 0.152, *p*-value = 0.66 | 0.993 |
| | | 3-1 (River discharge-Rainfall) | Beta kernel | | | | | | |
| | Tree 2 | 23\|1 | Bernstein | | 0.0005 | 0.0214 | 0.016 | D = 0.086, *p*-value = 0.99 | 0.992 |
| | | | Beta kernel | | | | | | |

Note: D-vine structure-2 (case-2) (bold letter with an asterisk) selected the most parsimonious vine density (exhibits the minimum value of MSE, RMSE MAE, K-S and high value of NSE test statistics).

Conditioning variables:
- Case 1: 3 or Maximum River Discharge (Time interval = ±4 days) placed in the centre
- Case 2: 2 or Maximum Storm Surge (Time interval = ±4 days) placed in the centre
- Case 3: 1 or Annual Maximum 24 h Rainfall placed in the centre

### 3.4. Comparing the Adequacy of Fitted Nonparametric D-Vine with Parametric and Semiparametric Approaches in the D-Vine Copula Framework

3.4.1. Constructing D-Vine Structure in the Parametric Fitting Procedure

In the parametric vine approach, at first, the best fitted univariate marginal pdfs, for instance, GEV (for rainfall), normal (for storm surge) and GEV (for river discharge) distribution, were selected (refer to Table 2). This was followed by the same steps we discussed in the last section. Three different D-vine structures (case 1, case 2 and case 3) were considered by permutating the conditioning variables (refer to Figure 2). Our previous study confirmed that Survival BB7 fit best for flood pair rain and river discharge, Survival BB1 for storm surge–river-discharge and BB1 copula for the rain and storm surge pair [63].

For vine structure 1 (case 1, river discharge as conditioning variable; refer to Figure 2 and Table 5), both the selected 2D copulas (Survival BB7 and Survival BB1) were employed in the estimation CCDFs, which became the input to define another 2D copula in the second tree level (Tree 1). The present study tested different parametric copulas to fit the D-vine structure's second tree level (Tree 2) (refer to Supplementary Table S3a–c). The parameters of the fitted copulas were estimated using maximum pseudo-likelihood estimation (MPL) [112,113], and the performances of the fitted models were compared using the Cramer–von Mises functional test statistics $S_n$, with the parametric bootstrap procedure (N is the number of bootstrap samples = 1000) [114,115]. From Table S3a–c, the Frank copula was identified as best for Tree 2 (D-vine structure 1, case 1), rotated BB6 270-degree copula for Tree 2 (D-vine structure 2, case 2) and Frank copula overall (D-vine structure 3, case 3). The full trivariate D-vine structure (parametric settings) for each case was obtained using Equation (15).

After developing vine structures for the given flood characteristics, their performances were compared to select the most efficient D-vine structure developed under parametric settings for three cases. From Table 5, it was found that D-vine structure-3 (case-3), with rainfall as a conditioning variable, outperformed other vine structures (minimum values of AIC, BIC, MSE, RMSE, MAE and K-S statistics and with high values of log-likelihood (L-L) and NSE statistics).

**Table 5.** Summary statistics of the fitted D-vine structure under parametric distribution settings (via parametric copulas with parametric marginal distributions).

| Vine Structure (Conditioning Variable) | Tree Level | Flood Attribute Pairs | Most Parsimonious Fitted Copula | Copula Dependence Parameters (θ) via MPL | MSE | RMSE | MAE | K-S | NSE | *Log-Likelihood (LL)* | *AIC* | *BIC* |
|---|---|---|---|---|---|---|---|---|---|---|---|---|
| Case 1 (1-3-2) | Tree 1 | 1-3 (Rain-River discharge) | Survival BB7 | $\theta$ = par = 1.142; $\delta$ = par2 = 0.19 | 0.00086 | 0.0294 | 0.021 | D = 0.152 (0.66) | 0.982 | 10.51 | −11.02 | −1.87 |
| | | 3-2 (storm surge-river discharge) | Survival BB1 | $\theta$ = par = 0.23; $\delta$ = par2 = 1.29 | | | | | | | | |
| 3 or Maximum River Discharge (Time interval = ±4 days) placed in the centre | Tree 2 | 12\|3 | Frank | $\theta$ = par = **2.92** | | | | | | | | |
| Case 2 (1-2-3) | Tree 1 | 1-2 (Rain–Storm Surge) | BB1 (Clayton-Gumbel) | $\theta$ = par = 0.19; $\delta$ = par2 = 1.36 | 0.00088 | 0.0297 | 0.022 | D = 0.173 (0.48) | 0.982 | 10.50 | −11.01 | −1.87 |
| | | 2-3 (Storm surge–river discharge) | Survival BB1 | $\theta$ = par = 0.23; $\delta$ = par2 = 1.29 | | | | | | | | |
| 2 or Maximum Storm Surge (Time interval = ±4 days) placed in the centre | Tree 2 | 13\|2 | Rotated BB6 270 degrees | $\theta$ = par = −1; $\delta$ = par2 = −1 | | | | | | | | |
| **Case 3 (2-1-3) \*** | Tree 1 | 2-1 (Storm surge-Rain) | BB1 (Clayton-Gumbel) | $\theta$ = par = 0.19; $\delta$ = par2 = 1.36 | **0.00084** | **0.0290** | **0.021** | **D = 0.152 (0.67)** | **0.982** | **10.85** | **−11.71** | **−2.57** |
| | | 3-1 (River discharge-Rainfall) | Survival BB7 | $\theta$ = par = 1.142; $\delta$ = par2 = 0.19 | | | | | | | | |
| 1 or Annual Maximum 24 h Rainfall placed in the centre | Tree 2 | 23\|1(Storm-River/Rainfall) | Frank | $\theta$ = par = 2.97 | | | | | | | | |

Note: D-vine structure-3 (case 3, indicated by bold letter with an asterisk), considering rainfall events as a conditioning variable, perform better (minimum value of AIC, BIC, HQC, MSE, RMSE, K-S and high value of model's L-L and NSE statistics).

### 3.4.2. Constructing a D-Vine Structure with the Semiparametric Settings

2D parametric class copulas were incorporated with the nonparametric marginal pdf in the semiparametric D-vine structure. Firstly, the best-fitted 2D parametric copulas for Tree-1 in all three cases of the D-vine structure (refer to Figure 2) were selected from our previous study [63]. Refer to Section 3.4.1. Supplementary Table S4a–c shows different parametric class 2D copulas fitted with an MPL-based parameter estimation procedure for estimating the most justifiable bivariate density fitted to the second tree level (Tree 2). The investigation found that the Frank copula was best for the D-vine structure-1 (case-1), the rotated BB6 270 degree one for vine structure-2 (case-2) and the Frank copula for D-vine structure-3 (case-3). Using Equation (15), the full trivariate vine copula joint density was estimated for each fitted D-vine structure. The most justifiable semiparametric-based vine structure was selected by comparing the performances of three different cases of D-vine structure. Table 6 provides the summary details of the fitted D-vine structures. It was found that D-vine structure 3 (case-3), considering rainfall as a conditioning variable, outperformed all other possible D-vine structures (case-1 and case-2); it exhibited minimum values of MSE, RMSE, MAE, K-S, AIC and BIC statistics and high values of model likelihood (L-L) and NSE statistics.

**Table 6.** Details of the fitted 3-D vine structure in the semiparametric distribution settings for three different cases.

| | Vine Structure (Conditioning Variable) | Tree Level | Flood Attribute Pairs | Most Parsimonious or Best-Fitted Copula | Copula Dependence Parameters (θ) | MSE | RMSE | MAE | K-S | NSE | Log-Likelihood (LL) | AIC (Akaike Information Criterion) | BIC (Bayesian Information Criterion) |
|---|---|---|---|---|---|---|---|---|---|---|---|---|---|
| Case 1 (1-3-2) | 3 or Maximum River Discharge (Time interval = ±4 Days) placed in the centre | Tree 1 | 1-3 (Rain-River discharge) | Survival BB7 | $\theta = $ par $= 1.142$; $\delta = $ par2 $= 0.19$ | 0.0006 | 0.0236 | 0.0185 | D = 0.152 (0.66) | 0.9886 | 10.68 | −11.37 | −2.23 |
| | | | 3-2 (storm surge-river discharge) | Survival BB1 | $\theta = $ par $= \mathbf{0.23}$; $\delta = $ par2 $= \mathbf{1.29}$ | | | | | | | | |
| | | Tree 2 | 12\|3 | Frank | $\theta = $ par $= \mathbf{2.89}$ | | | | | | | | |
| Case 2 (1-2-3) | 2 or Maximum Storm Surge (Time interval = ±4 days) placed in the center | Tree 1 | 1-2 (Rain–Storm Surge) | BB1 (Clayton-Gumbel) | $\theta = $ par $= 0.19$; $\delta = $ par2 $= 1.36$ | 0.0006 | 0.0239 | 0.0192 | D = 0.130 (0.82) | 0.9883 | 10.81 | −9.62 | 1.34 |
| | | | 2-3 (Storm surge–river discharge) | Survival BB1 | $\theta = $ par $= \mathbf{0.23}$; $\delta = $ par2 $= \mathbf{1.29}$ | | | | | | | | |
| | | Tree 2 | 13\|2 | Rotated BB6 270 degrees | $\theta = $ par $= -1$; $\delta = $ par2 $= -1$ | | | | | | | | |
| **Case 3 (2-1-3) \*** | 1 or Annual Maximum 24 h Rainfall placed in the center | Tree 1 | 2-1 (Storm surge-Rain) | BB1 (Clayton-Gumbel) | $\theta = $ par $= 0.19$; $\delta = $ par2 $= 1.360$ | **0.0005** | **0.0232** | **0.0185** | **D = 0.1303 (0.82)** | **0.9890** | **11.38** | **−12.77** | **−3.62** |
| | | | 3-1 (River discharge-Rainfall) | Survival BB7 | $\theta = $ par $= 1.14$; $\delta = $ par2 $= 0.19$ | | | | | | | | |
| | | Tree 2 | 23\|1 | Frank | $\theta = $ par $= 2.75$ | | | | | | | | |

Note: D-vine structure 3 (case 3) fitted best (bold letter with an asterisk) (minimum value of AIC, BIC, MSE, RMSE, MAE, K-S and high value of NSE and L-L statistics.

3.4.3. Comparison of the Models' Performances (Nonparametric vs. Semiparametric vs. Parametric Vine)

Section 3.3, Section 3.4.1, and Section 3.4.2 recognized the most justifiable vine structures, D-vine structure-2 (obtained via nonparametric vine approach), D-vine structure-3 (via parametric vine approach) and D-vine structure-3 (via semiparametric framework). We performed an analytical and graphical comparison to check the adequacy of the selected nonparametric D-vine density with parametric and semiparametric vine approaches fitted to the given triplet flood variables. Refer to Table 7. The D-vine structure (case-2) defined in the nonparametric setting outperformed the others (minimum values for MSE, RMSE, K-S, MAE and high NSE test statistics). It was also observed that the performance of the selected semiparametric D-vine structure (case-2) was better than that of the parametric D-vine structure (case-2) in the trivariate flood dependence. These results can further reveal how well the performance of the incorporated vine getting improves when switching its marginal distribution from parametric to non-parametric and the copula joint density from parametric to nonparametric joint pdf. The reliability and suitability of the selected D-vine structures were examined further by comparing Kendall's $\tau$ correlation coefficient estimated from the simulated flood events (sample size N = 1000) using the best-fitted nonparametric vine copula (D-vine structure-2), parametric vine (D-vine structure-2) and semiparametric vine copula (D-vine structure-3) and compared with the empirical Kendall's $\tau$ coefficient estimated from the historical flood events (refer to Table 8). It was found that the obtained nonparametric D-vine structure (case-2) exhibited a minimum gap or difference between the empirical and theoretical Kendall's tau statistics. These results confirm that the selected nonparametric vine structure regenerates the historical flood dependence structure (or correlation) much more efficiently. The same table also revealed that the semiparametric vine approach better captures and regenerates flood dependence than parametric vine copula density.

**Table 7.** Comparing the performance of the selected nonparametric D-vine structure with parametric and semiparametric vine copula density.

| Best-Fitted D-Vine Structure | MSE | RMSE | MAE | K-S | NSE |
|---|---|---|---|---|---|
| **Nonparametric settings (D-vine structure-2 (case-2) *** | **0.0002** | **0.0153** | **0.0130** | **D = 0.152, *p*-value = 0.66** | **0.995** |
| Semiparametric settings (D-vine structure-3 (case-3) | 0.0005 | 0.0232 | 0.0185 | D = 0.130 (0.82) | 0.989 |
| Parametric settings (D-vine structure-3 (case-3) | 0.00084 | 0.0290 | 0.0218 | D = 0.152 (0.67) | 0.982 |

Note: D-vine structure-2 derived in the nonparametric settings (bold letter with an asterisk) outperformed both parametric and semiparametric approaches in the D-vine structure for trivariate CF events.

A graphical visual inspection was carried out to crosscheck the adequacy of the selected D-vine structure-2 obtained nonparametrically. The overlapped scatterplots between the observed samples (via historical flood) and simulated samples (using D-vine structure-2, case-2) of sample size (N = 1000) were obtained; refer to Supplementary Figure S6a–c. It was found that D-vine structure-2 (under nonparametric settings) performs adequately since the simulated random sample (indicated by light grey colour) overlapped with the natural mutual concurrency of the historical flood samples (red colour); refer to Figure S6a–c. Supplementary Figure S7 illustrates a 3D scatterplot matrix of the generated flood events (sample size N = 1000) using the selected nonparametric vine model. Supplementary Figure S8 illustrates the vine tree structure of the most justifiable D-vine structure in the nonparametric setting.

**Table 8.** Examining the reliability of the developed D-vine structure (nonparametric settings) vs. parametric D-vine vs. semiparametric D-vine copula framework by comparing Kendall's τ correlation coefficient estimated using the generated random samples (size N = 1000) obtained from the above-selected model with Empirical Kendall's τ values estimated from historical observations.

| CF Pairs | Kendall's Tau Estimated from Historical Flood Events (Empirical Estimates) | Kendall's Tau Estimated from Best-Fitted D-Vine Copula (D-Vine Structure 3) in a Parametric Setting (Theoretical Estimates) | Kendall's Tau Calculated from Best-Fitted D-Vine Copula (D-Vine Structure-3) in a Semiparametric Setting (Theoretical Estimates) | Kendall's Tau Estimated from Best-Fitted D-Vine Copula (D-Vine Structure-2) in the Nonparametric Setting (Theoretical Estimates) |
|---|---|---|---|---|
| Annual maximum 24 h rainfall (mm) − Maximum Storm Surge (m) (Time interval = ±4 days) | 0.30 | 0.29 | 0.31 | 29 |
| Maximum Storm Surge (m) (Time interval = ±4 days) − Maximum River discharge (m³/s) (Time interval = ±4 days) | 0.29 | 0.34 | 0.31 | 29 |
| Annual maximum 24 h rainfall (mm) − Maximum Storm Surge (m) (Time interval = ±4 days) − Maximum River discharge (m³/s) (Time interval = ±4 days) | 0.14 | 0.16 | 0.15 | 0.14 |

Note: It is clearly found that the best-fitted D-vine structure-2 constructed in the nonparametric distribution setting are very much closer to its empirical estimates.

In conclusion, the above investigations confirm that the nonparametric vine density is much better in trivariate dependence analysis of CF events. This framework has no prior distributional assumption about their copula joint density and its marginal behaviour. Finally, the selected nonparametric vine density is employed to estimate trivariate joint cumulative distribution functions (JCDF) and joint return periods.

### 3.5. Compound Flooding Events Risk Assessments

Flood frequency analysis (FFA) establishes an interrelationship between the flood design quantiles and their non-exceedance probabilities by fitting the best-fitted univariate or multivariate probability distribution function. Primary return periods comprise two different joint cases, OR- and AND-joint. The fitted nonparametric D-vine structure was employed in estimating trivariate joint cumulative distribution function (JCDF) and trivariate return periods for OR- and AND-joint cases for different possible combinations of flood events; refer to Table 9 and Equations (16) and (17). Table 9 also shows estimations of the bivariate joint return periods using the best-fitted 2D nonparametric copula density (refer to Table 3). It was found that the trivariate return periods for the AND-joint case were higher than for the OR-joint case for the same flood combination. Similarly, the bivariate AND-joint case was higher than the OR-joint case for the same flood pair combinations. These results further reveal that the occurrence of trivariate flood events simultaneously is less frequent in the "AND" case and more frequent in the "OR" joint case. The same observations are also valid for the bivariate case. For instance, refer to Table 9: a 1-in-100-year flood event with the following characteristics—rainfall = 147.541 mm, storm surge = 0.337486 m and river discharge = 5951.523 $m^3s^{-1}$—the trivariate OR- and AND-joint return periods are 33.66 years and 3486.75 years. For the same flood characteristics mentioned above, the bivariate return periods for OR- and AND-joint cases are 50.92 years and 2744.23 years for flood pair rainfall and storm surge events; 50.26 years and 9633.91 years for storm surge and river discharge; and 50.29 years and 8517.88 years for rainfall and river discharge pair events.

**Table 9.** Comparing primary return periods (univariate vs. bivariate vs. trivariate) for a different possible combination of triplet flood events.

| Estimated Flood Quantiles Using the Inverse Cumulative Distribution Functions (CDFs) of Best-Fitted Marginal Distribution via KDE | | | | Bivariate Joint Return Periods (JRPs) | | | | | | Trivariate Joint Return Periods (JRPs) Estimated Using the Best-Fitted D-Vine Structure (Case-2) | |
|---|---|---|---|---|---|---|---|---|---|---|---|
| Return Period (Years), T | Annual Maximum 24 h Rainfall (R) (mm) | Maximum Storm Surge (m) (SS) (Time Interval = ±4 Days) | Maximum River Discharge (RD) ($m^3s^{-1}$) (Time Interval = ±4 Days)) | OR-JRP, $T_{RS}^{OR}$ | AND-JRP $T_{RS}^{AND}$ | OR-JRP, $T_{SRD}^{OR}$ | AND-JRP $T_{SRD}^{AND}$ | OR-JRP, $T_{RRD}^{OR}$ | AND-JRP, $T_{RRD}^{AND}$ | OR-JRP, $T_{RSRD}^{OR}$ | AND-JRP, $T_{RSRD}^{AND}$ |
| 5 | 97.04 | 0.151 | 2573.15 | 3.08 | 13.17 | 2.84 | 20.52 | 2.88 | 18.91 | 2.06 | 15.95 |
| 10 | 110.73 | 0.228 | 3367.54 | 5.69 | 40.97 | 5.32 | 81.46 | 5.34 | 77.23 | 3.70 | 50.42 |
| 20 | 128.04 | 0.277 | 5408.36 | 10.77 | 139.73 | 10.31 | 328.39 | 10.32 | 318.89 | 7.02 | 173.06 |
| 50 | 143.42 | 0.316 | 5854.64 | 25.85 | 760.16 | 25.29 | 2162.62 | 25.30 | 2063.98 | 17.00 | 950.29 |
| 100 | 147.54 | 0.337 | 5951.52 | 50.92 | 2744.23 | 50.26 | 9633.91 | 50.29 | 8517.88 | 33.66 | 3486.75 |

It is observed from the above-estimated return periods (refer to Table 9) that it would be preferable in practice to use trivariate return periods instead of the bivariate (or univariate). The above results also reveal that the accountability of both primary joint return periods is essential; just considering either AND-joint or OR-joint case would be problematic in the hydrologic risk evaluation. They also depend on the nature of water-related problems, which usually decide the importance of the different types of return periods.

In our current study, the developed nonparametric D-vine model was employed further in estimating the failure probability (FP) statistics. This risk approach examined variation in the trivariate, bivariate and univariate flood hazard events measured by service design lifetime for different return periods, such as 100 years, 50 years, 20 years, 10 years, and 5 years; refer to Figure 3a–e. A trivariate flood hazard scenario was found to result in higher-value FP than bivariate and univariate events. Both the trivariate and bivariate (also univariate) hydrologic risk or FP statistics are reduced when the return increases. FP statistics increase when there is an increase in the service design lifetime of the hydraulic infrastructure under consideration. For instance, at the return period of 100 years, the estimated value of FP statistics is 0.778 (for trivariate hazard scenario) and 0.629 (for bivariate hazard scenario) at a 50 year design lifetime. When considering a higher design lifetime, say 100 years, for the same return periods (100 years), the estimated value is 0.951 (for the trivariate hazard scenario) and 0.862 (for the bivariate scenario). Similarly, at 100-year return periods, the trivariate and bivariate hazard scenario is 0.933 and 0.832 (at 90 years design lifetime). When reducing the return periods to 50 years, the value is 0.995 and 0.971 at the same design lifetime (90 years).

From the above results, it is inferred that observing the joint behaviour of the storm surge event, rainfall event, and river discharge is essential in reducing the risk of coastal flood hazard. Considering the univariate probability analysis or even bivariate joint behaviour may underestimate the level of risk In conclusion, ignoring the trivariate probability analysis would be a problem which could result in the underestimation of FP. Their joint probability occurrence facilitates a better understanding and realization of extreme compound scenarios. All the above-discussed analytical and graphical investigations are crucial for sustainable design and planning in coastal flood management strategies.

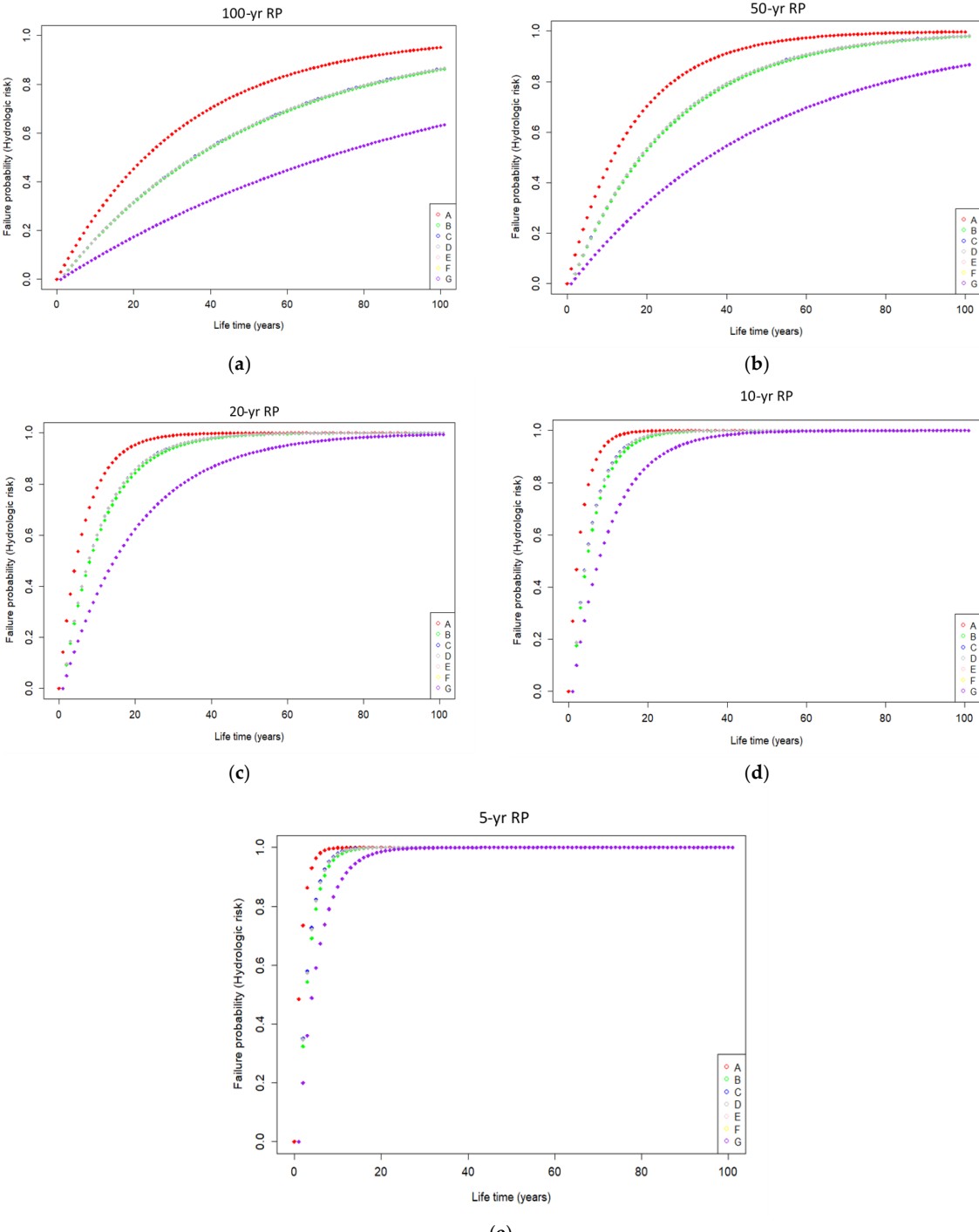

**Figure 3.** Assessments in the hydrologic risk of CF events for return periods (RPs) (**a**) 100 years, (**b**) 50 years, (**c**) 20 years (**d**) 10 years, (**e**) 5 years [Note: A [red colour]—describing trivariate CF hazard scenario for OR-joint case; B [green colour]—describing bivariate CF hazard scenario between flood pair rainfall and storm surge for OR-joint case; C [blue colour]—describing bivariate CF hazard scenario for flood pair storm surge and river discharge for OR-joint case; D [grey colour]—describing bivariate CF hazard scenario for flood pair rainfall and river discharge for OR-joint case; E [pink colour]—describing univariate hazard scenario through rainfall events; F [yellow colour]—describing univariate hazard scenario through storm surge events; G [purple colour]—describing univariate hazard scenario through river discharge events].

## 4. Research Summary and Conclusions

This study incorporated the D-vine copula in the nonparametric fitting procedure to model trivariate joint probability analysis of the storm surge, river discharge and rainfall in the compound flood risk assessments. The common forcing mechanisms that can derive multiple extreme events either successively or in close succession in the coastal regions can exacerbate the impact of flooding events. A comprehensive compound flood risk understanding can demand the accountability of multiple flood-driving agents simultaneously because the complex interplay between them can be devastating. The performance of the parametric and semiparametric approach in the vine framework was also compared with the proposed nonparametric vine density in the CF dependence. The traditional vine framework was defined by incorporating multiple parametric class 2D copula densities with parametric 1D univariate margins. This parametric density (and their marginal distribution) approximation had some statistical constraints already discussed in Section 1. The nonparametric via the Bernstein estimator and beta kernel copula estimator is a much more comprehensive way of vine construction, where the fitted 2D copula densities between each flood pair can adapt to any dependence structure without the requirement of any specific or fixed probability density structure. Conversely, the semiparametric vine framework integrated multiple 2D parametric class copulas with nonparametric marginal pdfs via the Kernel density estimation (KDE). The main findings of this study are summarized below:

1.  This study compounded the joint relationship between annual maximum 24 h rainfall and their associated storm surge and river discharge observed within a time lag of ±4 days from the date of highest rainfall events. Our previous study [63] already examined the degree of mutual concurrencies and confirmed a significant positive correlation between selected flood contributing-variables.

2.  Our previous study confirmed that rainfall and storm surge events did not exhibit any significant trend or serial correlation (or autocorrelation). From our present study [63], it was also confirmed that both variables are homogenous. However, the storm surge events exhibited nonstationary behaviour (time trend with non-homogeneity), but no serial correlation was identified.

3.  The nonparametric Normal KDE is selected as the most parsimonious for all three targeted flood variables (refer to Table 2a–c). Additionally, the results were the same when comparing the performance with the best-fitted parametric function (GEV, NORMAL, GEV [63]; refer to Table 2a–c. This further reveals that a lack of prior distributional assumption can result in a better explanation of the targeted flood marginal distribution behaviour.

4.  The 3-D vine copula was constructed by permutating the conditioning variable's location, which resulted in three different D-vine structures. In constructing the D-vine copula nonparametrically, it was found that the Bernstein copula fit best for flood pairs rainfall–storm surge and storm surge and river discharge, and the beta kernel estimator fit best for pair rainfall–river-discharge. All the selected nonparametric 2D copulas were employed in the 3D vine construction for three different D-vine structures. The fitness test statistics confirmed that nonparametric D-vine structure-2 (case-2) performs better when considering storm surge as a conditioning variable (with the Bernstein copula estimator fitted in both the first and second tree levels). It is important to note that it is much more practical to consider each targeted variable separately as a conditioning variable instead of just fixing it in the vine construction. This approach can generate multiple possible structures for selecting the most justifiable one.

5.  Similarly, the parametric and semiparametric vine fitting approaches selected D-vine structure-3 (rainfall as a conditioning variable; refer to Tables 5 and 6) as the most justifiable density based on different GOF test statistics.

6.  Best-fitted models have been compared analytically, such as nonparametric, semiparametric, and parametric fitting-based D-vine structures. Results confirmed the adequacy of the proposed nonparametric vine density. The model's reliability was

investigated analytically by comparing the theoretical and empirical Kendall's tau. Results revealed that the selected D-vine structure-2, in the nonparametric fitting procedure, outperformed the others. In other words, the selected vine structure regenerates historical flood events efficiently. The adequacy of D-vine structure-2 (Nonparametric framework) was further investigated graphically through overlapped scatterplots between historical observation and generated samples. It is clearly noted that the fitted model effectively captured the natural mutual dependencies of historical flood events. In conclusion, our proposed vine copula density in the nonparametric fitting is a much better alternative to the traditional parametric vine approach.

7. The best-fitted nonparametric vine density was employed to estimate trivariate primary joint return periods for OR- and AND-joint cases. The OR-joint case resulted in lower return periods than the AND-joint case for the same flood combinations. It was noted how important it is to take accountability for trivariate return periods rather than just considering a bivariate (or univariate) approach, which would be problematic and less efficient for solving different water-related decision-making.

8. The trivariate and bivariate joint CDFs were employed in estimating failure probability (FP) statistics which highlight the hydrologic risk due to the compound effect of rainfall, storm surge and river discharge events in the trivariate flood events. Investigation revealed that FP statistics could be underestimated if neglecting the trivariate joint probability analysis between targeted flood characteristics compared to when considering the same flood variables in pairwise joint modelling. The FP statistics were higher when considering trivariate joint distribution for the OR-joint event than when considering bivariate joint dependency between flood pairs. The hydrologic risk (trivariate, bivariate and univariate events) decreases with an increase in the return periods. At the same time, hydrologic risk increases, followed by the service design lifetime of hydraulic infrastructure under consideration. The same investigation also found that the FP of univariate flood events is much lower than trivariate (and bivariate) events. These further reveals that compound events may not be devastating if each flood source variable is considered separately.

This study has a few limitations. Firstly, this study only considered 46 years of the observational dataset. It might cause a source of uncertainty in the estimated outcomes. It could be preferred to take long-term data to reduce or minimize the risk of inheritance uncertainty. Secondly, our proposed model considers nonparametric distribution, both univariate marginals and multivariate nonparametric copula density in the vine construction. The model compatibility and performance have already been compared thoroughly with the existing parametric (or semiparametric) vine framework. It could be denied that there is much scope for applying this nonparametric framework to model the joint behaviour of different possible extreme events across the world. Even though it is possible to extend this proposed model to higher dimensional modelling for more than three variables. However, on the other side, it might not be easy to extrapolate to high return levels. It needs to be addressed in extreme event modelling. Our present study is not tackling this issue, which will be considered in our further study.

**Supplementary Materials:** The following supporting information can be downloaded at: https://www.mdpi.com/article/10.3390/hydrology9120221/s1.

**Author Contributions:** Project focus and supervision, S.P.S.; methodology, software, formal analysis, S.L.; writing—original draft preparation, S.L.; writing—review and editing, S.L. and S.P.S.; project administration, S.P.S.; funding acquisition, S.P.S. All authors have read and agreed to the published version of the manuscript.

**Funding:** This research was funded by the Natural Sciences and Engineering Research Council of Canada (NSERC) collaborative grant with the Institute for Catastrophic Loss Reduction (ICLR) to the second author [Collaborative Research and Development (CRD) Grant-CRDPJ 472152-14].

**Data Availability Statement:** Data used in the presented research are available at https://tides.gc.ca/eng/data (CWL data) (accessed on 9 June 2021).; https://wateroffice.ec.gc.ca/search/historical_e.html (Streamflow discharge records) (accessed on 15 June 2021); https://climate.weather.gc.ca/ (rainfall data) (accessed on 22 June 2021).

**Acknowledgments:** We are thankful for Canada's Fisheries and Ocean assistance for the coastal water level (CWL) data and Environment and Climate Change Canada for daily river discharge data. Special thanks to the Canadian Hydrographic Service (CHS) for providing the tide data. We are special thanks for the funding provided by NSERC (Natural Sciences and Engineering Research Council) and ICLR (Institute for Catastrophic Loss Reduction) in Canada.

**Conflicts of Interest:** The authors declare no conflict of interest.

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
