# Peer review of "Trivariate Joint Distribution Modelling of Compound Events Using the Nonparametric D-Vine Copula Developed Based on a Bernstein and Beta Kernel Copula Density Framework"

_hydrology, doi:10.3390/hydrology9120221_

Round 1

Reviewer 1 Report

Objectives can be presented (i).. (ii) .. 

The method and contributions are really sound, few lines on how the risk estimates can be conveyed to general public to be added (just before sect 4)

It is advisable to refer/add https://nhess.copernicus.org/articles/22/2145/2022/nhess-22-2145-2022-discussion.html

Reviewer 2 Report

The paper presents results of research on the compound flooding events risk assessments and the collective impact of the rainfall, storm surge and river discharge with the use of a non-parametric approach to constructing a 3D vine copula. The low-lying region at the Pacific coast and the Fraser River, Canada was taken as the case study. I found the article interesting. In my opinion, it is a valuable contribution to studies on the application of the copula function in the trivariate joint probability analyses of extreme hydro-meteorological (and oceanographic) events. This is a well-written paper, with an exhaustive description of methodology and an in-depth analysis of dependencies between the three variables. I don’t have critical comments to this part of the manuscript. However, I would suggest some corrections to the “Application” section:

1. “Our work introduces 46 years of selected flood characteristics (…)” (p. 10, l. 395), and then: “(…)) data were obtained of 1970 to 2018 from (…)” (p. 10, l. 403). It should be 49 years, not 46.

2. It would be helpful to the Readers to add a map showing the geographical position of the study area.

Moreover, editorial corrections related to the citation in the manuscript in accordance to the MDPI journal requirements, with reference numbers placed in square brackets, is necessary (please see Instructions for Authors).

To sum up, I recommend the paper for publication after minor amendments.

Reviewer 3 Report

Reviewer report – Manuscript “Trivariate joint distribution modelling of compound events using the nonparametric D-vine copula developed based on a Bernstein and Beta kernel copula density framework”

by Latif and Simonovic

The manuscript discusses the use of nonparametric statistics for the joint modeling of hydrological variables in Canada. The topic interesting and I feel the paper presents elements of novelty. However, there are some shortcomings that may prevent the publication of the paper in its present form. In fact, the manuscript is unnecessarily long, but fails to provide proper justification for some methodological steps and an in-depth discussion of the results. Also, there are several issues on the English writing that make reading and understanding difficult at some points. Hence, my recommendation is for major revision. What follows are general and specific comments, which I sincerely hope to be useful for the authors.

General comments:

1-   First and most important, a careful and thorough revision of the English writing must be performed by the authors. There are so many writing errors that it became unfeasible to point them out individually;

2-   The paper must be considerably shortened. In fact, several statements are consistently repeated along the manuscript whereas many others seem unrelated to the objectives of the study. I would also suggest some reorganization of the manuscript for clearly separating methodological aspects and results. This would certainly make reading easier and provide a more effective communication of the main findings of the study;

3-   While I understand the rationale of resorting to nonparametric statistics, I feel that the limitations of such an approach are not properly addressed by the authors. In fact, kernel density estimation (KDE) is usually a very flexible tool for probabilistic modeling (P6L259-260), but also presents serious limitations regarding extrapolation to high return levels, which, in my opinion, comprises the main objective of the study. Hence, despite the potential better fit to the available sample points, as compared to the parametric approach, the estimation of extreme quantiles based on KDE might be highly biased, particularly for environmental variables, which are, more often than not, heavy-tailed. The authors should elaborate this;

4-   Similar remarks can be made regarding the multivariate modeling. Copulas, even of the extreme type, are flexible enough for capturing dependence among variables, but might turn out to be asymptotically independent, which would hinder the estimation of joint return periods and/or risk. This shortcoming might be greatly enhanced when the fully nonparametric approach is utilized. The authors should elaborate this;

5-   P11L440-441: to the best of my knowledge, pre-whitening is utilized for removing serial correlation, not trends. Please provide some references that the technique is properly utilized. Also, I would understand using some kind of trend removing approach for inference purposes, but I am not sure that meaningful estimation of return levels is possible when stationary and nonstationary processes are combined for joint modeling of hydrological variables. I might be missing something here;

6-   Results should always be discussed, not only presented. How do the authors’ findings relate to previous studies, particularly regarding the joint risk? I think an in-depth discussion is missing;

7-   The conclusion section does not present the conclusions per se, but mostly repeat some of the results. I would suggest the entire section be rewritten, highlighting the actual findings of the study and the envisaged research developments.

Specific comments:

1-      A more focused literature review could be provided by the authors. The research gaps might be more clearly presented;

2-      P2L94 – what do the authors mean by “lower stages dependencies”. Employing clear and correct terminology is paramount for the proper understanding of the paper;

3-      P3L124 – I agree with these statements, but hydrological variables are rarely multimodal and a broad variety of theoretical distributions is flexible enough for modeling skewed variables. I would suggest reformulating these arguments;

4-      P3L128-130 – why “fixing the joint PDF of the dependence structure to any specific or predefined copula class must fail to fully acknowledge the flexibility of the copula fitted in the vine tree structure”? This is not always the case. The authors should temper this comment (and several similar ones along the manuscript);

5-      P3L131-133: how would a “time consuming” estimation procedure “contribute to underestimating the actual joint probability density”?

6-      P4L153 – the study is the first to the authors knowledge;

7-      Equation 2 – I would suggest replacing “f” and “g” for other symbols. These letters are usually used for denoting functions;

8-      P6L260 – Again, the nonparametric approach is flexible, but does not always performs better than the parametric counterpart, mainly when the interest lies in extrapolation;

9-      Please check Equation 12;

10-  P9L333 – what do the authors mean by “flood marginals”?

11-  P9L359 – what do the authors mean by “recursion interval”?

12-  Please check the notation in Equation 17;

13-  Section 3.1 – a few more details on the study area and on the utilized data within the manuscript would be useful;

14-  P10L395 – what do the authors mean by “flood characteristics”? Are the referring to the variables utilized in the study?

15- P10L401 – is there some justification for the time window of  days?

16-  P10L407 – what do the authors mean by “high astronomical data”?

17-  P11L423 – normality or normal distribution?

18-  Tables 5 and 6 – remove the “names” of the Greek letters (inside brackets);

19-  P21L634 – what do the authors mean by “regenerates the historical flood dependence structure”?

20-  P22L655 – the nonparametric approach might be “better” for reproducing the observed sample points due to the high flexible, but the same will not probably hold for extrapolation;

21-  P23L690 – the concept of “failure probability” should be presented and explained in the methodology, not along with the results;

22-  P23L702 – isn’t this result expected?

23-  P24L711 – are these values correct? Please check.

Round 2

Reviewer 3 Report

The manuscript has improved and at least some of the points raised in my previous review were properly addressed by the authors. However, there are still many issues in the English writing (the paper should preferably be revised by a native English speaker), and I think that a critical evaluation of the results must be provided. Hence, my recommendation is returning the paper for the authors for a minor revision.
